# The Effect of a Structured Intervention to Improve Motor Skills in Preschool Children: Results of a Randomized Controlled Trial Nested in a Cohort Study of Danish Preschool Children, the MiPS Study

**DOI:** 10.3390/ijerph182312272

**Published:** 2021-11-23

**Authors:** Lise Hestbaek, Werner Vach, Sarah Thurøe Andersen, Henrik Hein Lauridsen

**Affiliations:** 1Chiropractic Knowledge Hub, University of Southern Denmark, Campusvej 55, 5230 Odense, Denmark; 2Department of Sports Science and Clinical Biomechanics, University of Southern Denmark, Campusvej 55, 5230 Odense, Denmark; sthuroee@health.sdu.dk (S.T.A.); hlauridsen@health.sdu.dk (H.H.L.); 3Basel Academy for Quality and Research in Medicine, Steinenring 6, CH-4051 Basel, Switzerland; werner.vach@basel-academy.ch

**Keywords:** children, motor skills, preschool, kindergarten, randomized controlled trial

## Abstract

The preschool age seems to be decisive for the development of motor skills and probably the most promising time-window in relation to improvement of motor skills. This trial investigates the effect of an intensive structured intervention to improve motor skills in 3–6-year-old preschool children. A total of 471 Danish preschool children participated in a cluster randomized controlled trial. The intervention was to enhance motor skills, including predefined minimum criteria. Motor skills were measured using the Motor Assessment Battery for Children-2 (MABC-2) (total and three domains) at baseline and 6-, 18-, and 30-months post-intervention. The effect was estimated by multilevel linear regression with preschool and child included as random effects and adjusted for baseline values. Effect estimates were mainly positive at 6 months, but negative at 30 months with very few statistically significant estimates. In preschools with baseline motor skills scores below average, there was a tendency towards a larger improvement in the intervention group. Future interventions and research should focus on clusters with poor motor skills, as there is larger room for improvement. It remains to be seen whether the intervention can influence general well-being, physical activity, and self-perceived competence, both short- and long-term.

## 1. Background

Good motor skills are considered important for children’s physical, social, and psychological development [1] and may be the foundation for an active lifestyle, since motor skills form the building blocks for more complex and specialized skills essential for participation in physical activities [2] and thus, several studies have shown a positive association between good motor skills and higher levels of physical activity [2,3,4]. It has even been shown that motor skills proficiency at age 6 was strongly associated with physical activity at age 26 [5]. Either because of or besides the associations with physical activity, there is evidence of many health benefits to be gained from an improvement in motor skills. For instance, it has been demonstrated that good motor skills positively influence cardiorespiratory fitness [2,6], body weight [2,7,8,9], as well as sports participation [2,10], all suggesting that early competency in motor skills may have important health implications. Furthermore, there are indications of relationships with language development [11,12,13,14,15], executive function [16], and general wellbeing [17].

There are indications that the relative level of motor skills for age remains stable over time [18] and motor development deficits observed in early childhood are still apparent in adolescence [19]. Therefore, the toddler and preschool years appear to be particularly important periods for the development of motor skills that do not develop naturally, but are learnt, practiced, and developed over time [20]. Early childhood is thus the age where practicing fundamental movement skills is necessary to create a foundation for more complex movement activities of daily living, recreation, and sports in later childhood [1].

Despite the obvious importance of the development of motor skills in the preschool age, there are very few high-quality trials, i.e., randomized trials of sufficient size, investigating the effectiveness of motor skills interventions in normal 3–5-year-old children, whereas there has been more focus on physical activity or on motor skills in children with disabilities or in school aged children [21,22,23,24].

In Denmark, 92% of all 3–5-year old children spend a high proportion of their waking hours in preschool [25]. Pre-school education in Denmark is voluntary and takes place in different types of schools or day care centers covering the time before children enter compulsory education. A total of 75% of established day-care institutions are municipal day-care centers, while the other 25% are privately owned and are run by associations, parents, or businesses in agreement with local authorities. In terms of both finances and subject-matter, municipal and private institutions function according to the same principles and thus, this arena provides an ideal opportunity for all children, regardless of socioeconomic background, to develop and improve their motor skills.

Consequently, Danish preschools have had an increasingly strong focus on improving children’s motor skills. Several projects have been implemented both by municipalities and by larger sports organizations. For example, Sport Confederation Denmark has developed a program to improve motor skills in preschool children in cooperation with the Young Men’s Christian Association (YMCA), which is currently being offered to more than 40,000 children across Denmark [26]. Therefore, it seems timely to investigate the effectiveness and potential benefits of such programs.

Understanding the importance, but also the challenge, of early prevention, the Municipality of Svendborg in Southern Denmark has initiated an intervention in its preschools aimed at improving motor skills in preschool children. Furthermore, a partnership was established with the University of Southern Denmark to perform a scientific evaluation of the process and the effect. The municipality is comparable to the rest of Denmark in terms of age distribution, gender and income, but with a slightly higher unemployment rate (5.3% vs. 4.5%) [25], and therefore the results will be transferable to the rest of Denmark, provided other relevant factors such as outdoor areas, staffing, etc., are comparable.

The objective of this study is to investigate whether a structured program to promote motor skills in 3–6-year-old children can improve the children’s motor skills over a course of 30 months.

## 2. Methods

### 2.1. Design

This is a cluster Randomized Controlled Trial (RCT) in a natural municipal setting.

### 2.2. Setting

The trial is part of the Motor skills in Preschool Study (The MiPS study), where a cluster randomized controlled trial is nested within a larger cohort study.

Extensive testing of the children at an early age forms the basis of a cohort with potential for long-term follow-up, enabling investigations into the long-term development of motor skills, musculoskeletal disorders, physical activity, language, cognitive abilities, and social skills as well as the interrelations between these domains. In addition, the predictive ability of early markers for child development and health within these domains can be assessed. Thus, in addition to assessing the effectiveness of the intervention developed for this project, an evidence base for future strategies for optimizing children’s health, wellbeing and cognitive and language development has been established [27]. This article focuses explicitly on the effect of the intervention with regard to motor skills.

### 2.3. Participants

#### 2.3.1. Participating Preschools

All 31 preschools of the Municipality of Svendborg were included in the cohort and offered participation in the RCT. The governing boards of the preschools decided whether to partake in the RCT.

#### 2.3.2. Recruitment of Children

All parents of children attending public preschools in the Municipality of Svendborg (1461 children August 2016; 84% of the population in the age group) were invited to participate in the MIPs study. All parents received written information about the project and were invited to information meetings at local schools or preschools during the spring of 2016. Written consent forms, including information about the RCT, were signed, and returned to the children’s respective preschools before 1 September 2016.

The governing boards of 17 of the preschools (representing 834 children in August 2016) agreed to take part in the RCT.

Following the initial inclusion, running inclusion continued until the end of January 2017, at which point collection of baseline data was completed.

### 2.4. Randomization

Participating preschools were cluster-randomized, stratified for average socioeconomic background. A socioeconomic background index was defined for each preschool based on family type (single vs. two parents in the household), gross household income for 2015, and parental education (highest education for both parents) in the uptake area of the respective preschools. This information was available from the municipality and two strata were constructed: one with socioeconomic index above the median and one with an index below the median. This was done by a statistician at the University of Southern Denmark.

### 2.5. Follow Up

For all children, a follow up after 6, 18, and 30 months was planned. However, this was restricted to children who still attended preschool at the time of follow up. Each summer, the oldest year-group left the preschools to start school, which explains the majority of ‘loss to follow up’. In addition, a few children moved away from the municipality or to a non-participating preschool and were excluded from analyses. Children moving to another preschool within the same participation group were kept in the analyses.

### 2.6. Data Collection Methods

Trained research staff collected data from physical tests, including measurements of motor skills, physical competency, anthropometry and movement patterns. These tests were conducted at baseline (September 2016 to January 2017) and after 6, 18, and 30 months in sports arenas or gymnasiums in the proximity of the preschools. The children walked or were transported in buses, depending on the distance, and were accompanied by their known preschool staff.

Data were entered directly into REDCap [28] and stored at a secure database [29]. This manuscript only relates to motor skills. Results regarding the other domains will be presented in later articles to come.

### 2.7. Intervention

The intervention was developed by a working group that included representatives from the research team, the participating preschools, the Municipality and independent experts, in an iterative process aimed at ensuring ‘buy-in’ and ‘ownership’ of all stakeholders. The focus was on movement, development of motor skills, and body awareness.

Through the collaborative partnership with the individual preschool institutions, the intervention was designed to be flexible and adaptive to ease implementation. Researchers and independent experts provided examples and targets for motor skills, whereafter preschool staff modified the intervention strategies to suit the individual institutions’ physical environment, culture, and daily schedule. Hence, the intervention was not a strictly defined curriculum, but a strategy to enhance motor skills during the preschool day while fulfilling defined minimum criteria as described in Table 1. In addition, the majority of the adult-led activities should be carried out outdoors, and all children were required to have an elevated pulse every day.

### 2.8. Implementation

To support the implementation of the concept, a network of coordinators was established from the participating children’s institutions, and regular network meetings where these coordinators explain and discuss the implementation in their respective institutions were conducted. This offered the opportunity to solve any challenges along the way and provided mutual inspiration. Networks are important tools to support implementation and overcome the challenges that may arise during implementation [30]. Furthermore, the coordinators and preschool managers maintained ongoing evaluation schemes, e.g., periods where the amount of time spent in teacher-led activity will be monitored, or regular discussions about whether all three domains mentioned in Table 1 have been covered. This was to uncover potential implementation challenges needing to be addressed.

The description of the project, as developed by the administration in the municipality, can be seen in Appendix A in Appendix A, including further details with respect to implementation.

The implementation process was monitored and assessed continuously by the research team [31,32].

### 2.9. Competency Development

Svendborg Municipality recognized that the requirements for effective integration of the overall motor skills program into the daily practice of the institutions were considerable, and that not all staff members could contribute off hand. Therefore, funds were allocated to the preparation of a competency development program, supplying enrolled staff with tailored knowledge, skills and capacity to deliver the program. For this purpose, resources for 37 hours of training for all preschool staff in the participating preschools were allocated.

Preschool leaders and staff were represented in the working group for competency development, along with a municipal physiotherapist, a health care consultant, and others. Additionally, training was conducted in cooperation with the regional University College, which is responsible for the educations in physiotherapy and pedagogics.

The intervention was gradually introduced in the intervention preschools from September 2016 to January 2017.

### 2.10. Variables

#### Objectively Measured Motor Skills

Motor skills were tested with the revised version of the Movement Assessment Battery for Children (MABC-2) (Pearson, London, UK). The battery assesses the developmental status of fundamental movement skills in children and includes eight individual test items measuring movement skills in three domains: manual dexterity skills, ball skills and balance skills.

The tests included in the three domains, were as follows:
Manual dexterity: a one-hand posting task (Posting Coins), a timed bimanual task (Threading Beads), and an untimed drawing task (Drawing Trail).Ball skills: throwing an object to a target (Throwing Beanbag onto Mat) and catching an object using both hands (Catching Beanbag).Balance skills: a static balance task (One-leg Balance) and two dynamic tasks involving sustained, controlled movement (Walking Heels Raised), and more explosive action (Jumping on Mat).


The revised version also has qualitative assessments added, but only the quantitative assessments will be used for this study. The test (both the original and the revised) has been validated in several countries [33,34,35,36,37] and translated into Danish and has been widely used in Denmark [38]. The cross-cultural validation, the availability in several European countries and the simple test administration, facilitating large sample screening over a short period, are considered as major advantages of this test [39].

A total score expresses the child’s test performance and domain scores provide information about the child’s performance of each of the three domains [40]. The revised version of the test is subdivided into three age bands (3 to 6, 7 to 10, and 11 to 16) [41], thus the first age band will be used in this study. To allow comparison across age groups and across domains, the domain scores as well as the total scores are standardized as described in the manual, using the original United Kingdom norms [41], with the standardized scores ranging from 1 to 19.

### 2.11. Trial Registration

Registered 13 October 2016 in the ISRCTN registry; ISRCTN23701994

## 3. Covariates

Age was calculated in years at the day of the tests. Weight, height, and waist circumference were measured by the research staff at all follow-up points.

Socioeconomic variables were reported by the parents in a questionnaire distributed by mail at baseline: Mother’s and father’s highest completed education categorized into three categories: (1) secondary school or less (early childhood education, primary education, lower secondary education, upper secondary education); (2) vocational or short education (post-secondary non-tertiary education, short-cycle tertiary education, bachelor’s or equivalent level; i.e., ≤3 years after high school); and (3) academic education (Master’s or equivalent level, post-doctoral or equivalent level, i.e., >3 years) [42]. Family income (reported on a scale from 1 (DKK <200,000 in 2015) to 8 (DKK ≥800,000 in 2015) with increments of 100,000) was converted into equivalized household income. For this purpose, household members were converted into equivalized adults according to their age, using the so-called modified OECD equivalence scale (1.0 to the first adult; 0.5 to the second and each subsequent person aged 14 and over; 0.3 to each child aged under 14) [43]. The total income of a household was then divided by the converted number of household members.

### Sample Size Calculation

We are not aware of any previous studies using the same test in the same age groups, and therefore a power calculation for the RCT was performed using data from related studies. The standard deviations of the change in motor performance was based on a study of Danish overweight children [38], which used the same motor assessment (Movement Assessment Battery for Children), and the clustering effect was estimated based on a previous study in Danish preschools [44].

Based on the recommendations by J. In, the effect size was expressed as standard deviations units, since exact prior knowledge is not available. He states: “When prior knowledge for the calculation of the standardized effect size is not sufficient, a commonly applied effect size is 0.25–0.50, which was initially suggested by Cohen and which is still important.” [45]. Hence, we intended to recruit a sample large enough to detect a reduction in the standard deviation score of 0.30 with a power of 80% at a significance level of 5%. Correlation within preschools and hence the loss of efficiency because of clustering was accounted for by adjusting for intraclass correlation in the primary sampling units (ICC = 0.015). This power calculation resulted in a required number of 468 participants. Such a calculation should be considered with caution since the underlying estimates of variance and cluster effects are not directly transferable. However, the population size seems sensible in comparison to other trials investigating motor activities in preschools. Piek et al. included almost the same, 501 children [46], whereas Roth et al. investigated the effect of a physical activity intervention in preschools in a somewhat larger sample of 709 children [47]. However, many studies have demonstrated statistically significant effects with considerably smaller samples [48,49,50,51].

## 4. Analyses

As we have no information about the children not participating in the cohort, comparison between participants and non-participants at the study level cannot be performed. However, comparison between children in the cohort from preschools participating or not participating in the RCT with respect to basic demographics was conducted and the statistical significance of differences assessed by means of Pearson’s Chi^2^ or *t*-test.

Simple univariate statistics were used to describe the outcomes and covariates by intervention group. Statistical significance was assessed by corresponding regression models, taking the clustering within preschools into account by applying Stata’s cluster option.

Multilevel linear regression models were constructed to assess the effectiveness of the intervention on motor skills performance. The outcomes at follow up were modeled as a function of the baseline values of the outcome variable, time point of measurement, and time-point specific intervention effects. Any imbalanced covariate should be added to the model as fixed effects along with preschool, individual intercept and individual slope as random effects, allowing also a correlation between intercept and slope.

The main purpose of any public health intervention is to decrease the risk of subsequent problems for the most vulnerable group, in this case individuals with poor motor skills at baseline, rather than the strongest individuals. Therefore, the above analyses will be repeated on a subsample of children considered to be at risk of motor deficiency, i.e., scores below the 15% percentile at baseline, for both the individual domains and the total score [52].

Furthermore, prior to the study, there were variations among the preschools regarding pedagogical focus, where some preschools were more focused on physical and outdoor activities, others on developing social competencies, etc. Thus, the need for improvement of the children’s motor skills were expected to differ, and some preschools were expected to fulfill the requirements in the intervention already prior to the study, rendering additional improvement unlikely. Therefore, the analyses of effect were repeated on the subsample of preschools with baseline values of MABC-2 below the mean for both the individual domains and the total score. To illustrate the importance of baseline levels by preschool, scatter plots of change scores vs. baseline values for each child, stratified by preschool were constructed.

## 5. Results

Of the 834 children invited to participate in the RCT, parental consent was received from 471 children (56%), which were included in the RCT and available for testing at baseline and at 6 months follow-up, 259 children were available for testing at the 18 months follow-up, and 89 for the 30 months follow-up. The participation rates among these were 97%, 92%, and 97%, respectively, for the 6-, 18-, and 30-months follow-ups (Figure 1).

Sociodemographic baseline differences between participants and non-participants in the RCT are described in Table 2, showing a slightly higher educational level among the fathers in the non-RCT group than among participants, but no differences in sex, age, household income, or mother’s education.

Population comparisons between baseline and the three follow-up times are shown in Table 3. The dominant reason for non-participation at follow-up is leaving preschool to attend school (Figure 1) Therefore, the follow-up populations are slightly younger, and thus also smaller, than the baseline population, as children born late in the year are more likely to have school start postponed. They also have slightly lower MABC-2 scores.

### 5.1. Descriptive Analyses

Both demographic, anthropometric, and motor skills variables are presented by intervention group for baseline and all three follow-ups in Table 3. There were no differences between the intervention and the control group for any of the demographic or anthropometric variables, neither at baseline nor follow-up (Table 4). The intervention group improved more than the control group during the first six months, and children scores higher in the intervention group after 6 months. However, over time the picture reversed, and at the 30 months follow-up, all measurements were better in the control group.

### 5.2. Change Scores

The change scores from baseline to follow-up are presented in Table 5. This also demonstrates a larger improvement in the intervention group over the first six months which disappears over time, except for aiming and catching. When adjusted for the cluster effect of preschool, two of the four differences between groups are statistically significant at six months follow-up, whereas there are no statistically significant changes at the other two follow-ups except for the domain score for aiming and catching at 18 months follow-up.

### 5.3. Multilevel Models

None of the covariates were imbalanced between the two groups at baseline (Table 3), and thus none were included in the models. The multilevel analyses demonstrated statistically significant effects of the intervention at specific timepoints for specific outcomes. As in the bivariate models, the direction of effect was in favor of the intervention group at 6 months, mixed at 18 months, and the control group at 30 months (Table 6). Figure 2 illustrates the change over time, based on the baseline level followed by the predicted level of motor skills for each follow-up timepoint. It is seen that there is a bigger increase in the intervention group than in the control group over the first six months (from 2016 to 2017) whereafter the control group catches up, or even passes the intervention group.

The change scores for the total MABC-2 for each individual child in relation to their baseline scores are illustrated in Figure 3. There is a subfigure for each preschool. This figure illustrates that children with lower baseline values of the total MABC-2 score tend to show greater improvement, visualized by the regression lines. This holds both for children from the intervention group and from the control group, but the relation seems to be more pronounced in the intervention schools. Moreover, we can see a tendency that preschools with the lowest mean scores at baseline tend to show the highest mean change scores. Graphs for the individual domains demonstrate similar patterns (shown in Appendix A in Appendix A).

When isolating the children at high risk of motor development deficiency, the sample size was too small for meaningful modelling (*n* = 63, 34 and 13 for the 6-, 18-, and 30-months follow-ups, respectively), but the direction of effect were similar as for the whole sample (Table 6).

When repeating the multilevel regression analyses without the preschools who did well before the start of the intervention, i.e., the preschools with mean scores above the median, the results favored the intervention group for all domains and at all follow-ups except for manual dexterity at 18 months and aiming and catching at 30 months. However, the sample size is severely diminished, the confidence intervals very wide and the result only statistically significant in four of the twelve analyses (Table 6).

## 6. Discussion

This study demonstrated larger improvement in terms of motor skills over the first 6 months in the intervention group than in the control group. However, the effect diminished over time, and after 30 months the control group had slightly better motor skills than the intervention group. Children with low baseline values have the greatest improvement potential and when isolating the preschools with motor skills scores below average, there was a tendency towards a larger improvement in the intervention group than in the control group at all three follow-up times, but the sample was too small to show statistical significance. Interestingly, the same tendency was not seen when isolating individuals at high risk of motor deficiency, indicating that the effect is at preschool level rather than individual level. It is possible that the preschools with better motor skills scores at baseline were already doing well in stimulating the children’s motor skills and therefore had less improvement potential.

Most studies of motor skills interventions only included pre-post intervention comparisons and/or short-term analyses and reported positive results [47,49]. A metanalysis from 2017 demonstrated statistically significant effects for 25 RCTs investigating the effect of interventions to improve fundamental motor skills in the same age group as considered in this study. As in the present study, they found a distinctly smaller effect in studies with long term follow-up (≥6 months) than for shorter follow up times [24]. However, the metanalysis is not directly comparable to our results, since the included RCTs all implemented time-limited interventions, with follow-up after completion of the intervention. The intervention in our study continued throughout the children’s time in preschool, i.e., the follow-up period.

There are several possible explanations why the effect of the intervention appears to decrease over time in our study. One possible explanation is that the staff’s enthusiasm decreases with time. However, evaluation of the implementation indicated that the staff was happy with the program and continued the intervention for at least two years [31]. It is of course possible that this evaluation has not captured a possible decrease in the intensity of applying the intervention among the staff. Another possible explanation could be that the intervention accelerated the development of the children’s motor skills until they reached a certain degree of saturation, whereafter the control group caught up later, just at a slower pace. If this is the case, the intervention is probably not cost-effective. Finally, a contamination effect cannot be ruled out, as the intervention continued over a long period of time within the same municipality, where there is an ongoing contact between institutions. It is possible that the focus on motor skills throughout the municipality has influenced the staff in the control schools, and that the effect of this will materialize gradually over time. If this is the case, the intervention might still be efficient.

Some studies of motor skills intervention focus on children with delayed motor development and report positive results (e.g., Bardid et al. [49]). A review by Kirk et al. demonstrated considerably larger effects for children at risk of delayed motor skills than for normally developing children [53]. This is in line with our results, where effect sizes were larger for preschools with a generally low level of motor skills at baseline. However, we did not demonstrate better results for individual children with a low level of motor skills, which probably reflects that the intervention was not targeted to these children but implemented at group level.

A study from Australia closely resembled the present study where an intervention to improve motor skills in children at preschool age was developed and evaluated over an 18-month period. They found slightly better effects than in the present study which could be due to the fact that they only included schools from low socio-economic areas which has been associated with poor motor skills [54]. Therefore, the sample might resemble our preschools with baseline values below the median rather than our entire sample in which case, results are comparable. However, the intervention effects were not controlled for baseline values, and the baseline levels were generally lower for the intervention group. This may also explain the quantitative differences compared to our study.

The implications for practice and policy are not unambiguous. The results from the present study combined with the previous literature can encourage implementation of interventions to improve children’s motor skills in preschools where the level of motor skills is below average. However, if the goal is to improve the motor skills of the few individuals with poor baseline skills, the intervention did not appear efficient and thus it may be worth to consider targeting these children individually.

It cannot be ruled out that implementation was not optimal—at least in some preschools. However, when implementing an intervention in a real-world setting, it is difficult to imagine larger efforts than those displayed in the MiPS-study. The municipality provided a full week of education for all preschool staff, the intervention was developed in an iterative process between staff and experts, and network possibilities were established. As mentioned above, also the evaluation of the implementation indicated no lack of efficiency—nor in the long run.

There may be some concern about the disparity between the skills targeted in the intervention and the motor skills domains measured by the MABC-2. Including other measures of motor skills may have resulted in a different picture although we consider this unlikely, because the skills assessed with the MABC-2 are rather comprehensive for general motor competency. Even if some detailed test of a specific motor skill could potentially show a more distinct intervention effect, this would not be very relevant if the more common domains were not affected.

Other domains not directly related to motor skills might be influenced by the intervention. As the intervention includes both social and body awareness elements, among others, it is possible that the intervention can influence the general well-being, the amount of physical activity, or the subjective perception of own skills. Importantly, a child’s self-perceived competence has been shown to be crucial when understanding reduced participation patterns [55,56]. It is known that if children are confident about their motor proficiency, they are more likely to engage in activities such as sports, crafts, dance, and other physical activity programs outside of the school curriculum, which are also important for psychosocial development [57]. This suggests that targeting motor proficiency may in turn improve a child’s sense of self, and ultimately positively impact participation levels and overall social and emotional well-being. Further research is necessary to investigate whether the intervention considered may have a long-term effect on confidence, physical activity, etc.

## 7. Conclusions

Introducing a comprehensive intervention with focus on improvement of motor skills in Danish preschools improved the children’s motor skills as measured by MABC-2 in the short term. To draw firm conclusions about long term effects, further research must be conducted to illuminate the mechanisms behind the decrease in effect over time observed in this study.

When isolating the preschools with motor skills scores below average, there was a tendency towards a larger improvement in the intervention group than in the control group, both short- and long-term, but the sample was too small to show statistical significance. Thus, a tentative recommendation is to introduce measures to improve motor skills in institutions with below average baseline values.

Future studies should take baseline values of schools or preschools into account already in the design phase. It remains to be investigated whether the intervention can influence also general well-being, physical activity, and self-perceived competence, both short- and long-term.

## Figures and Tables

**Figure 1 ijerph-18-12272-f001:**
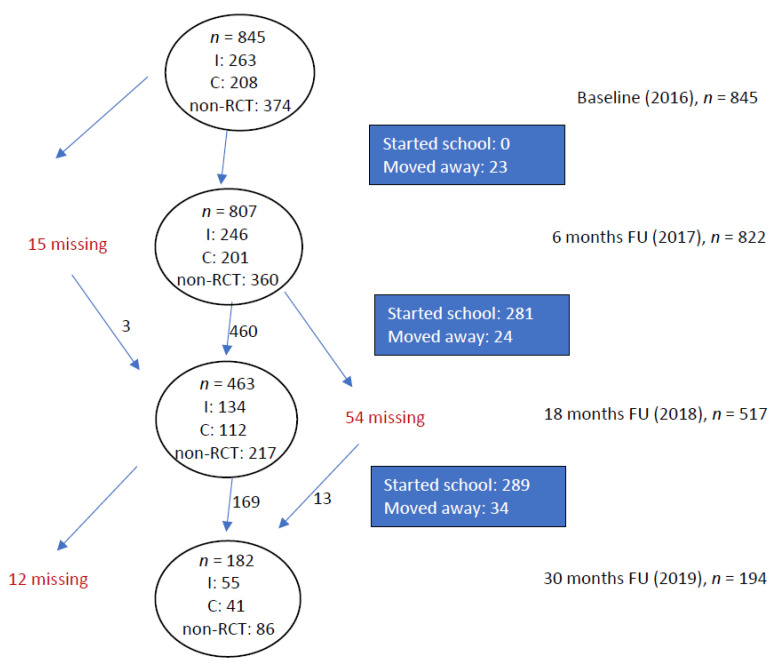
Movement ABC-2 measurements for the full cohort, including distributions in intervention groups. I: Intervention group; C: Control group; non-RCT: participating in the cohort but not the RCT; FU: Follow-up.

**Figure 2 ijerph-18-12272-f002:**
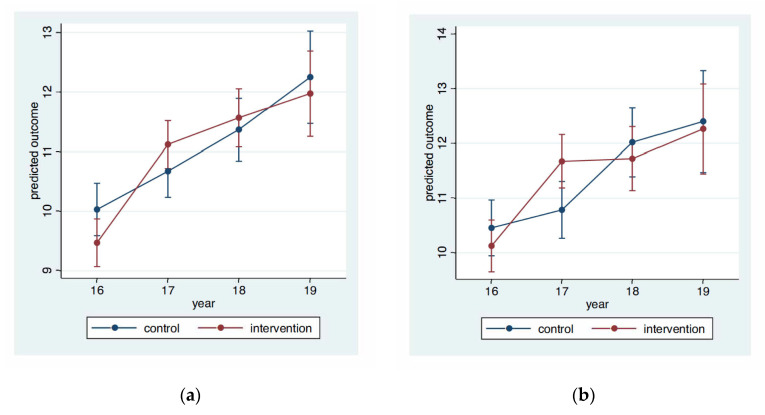
Predicted outcomes of MABC-2 with 95% confidence intervals at each time point for the two groups. (**a**) depict the total score, (**b**) the manual dexterity domain, (**c**) the balance domain, and (**d**) the domain for aiming and catching. Predictions are based on a model with the same random effects structure as the model used in the main analysis, but use the baseline values as outcome and not as covariate.

**Figure 3 ijerph-18-12272-f003:**
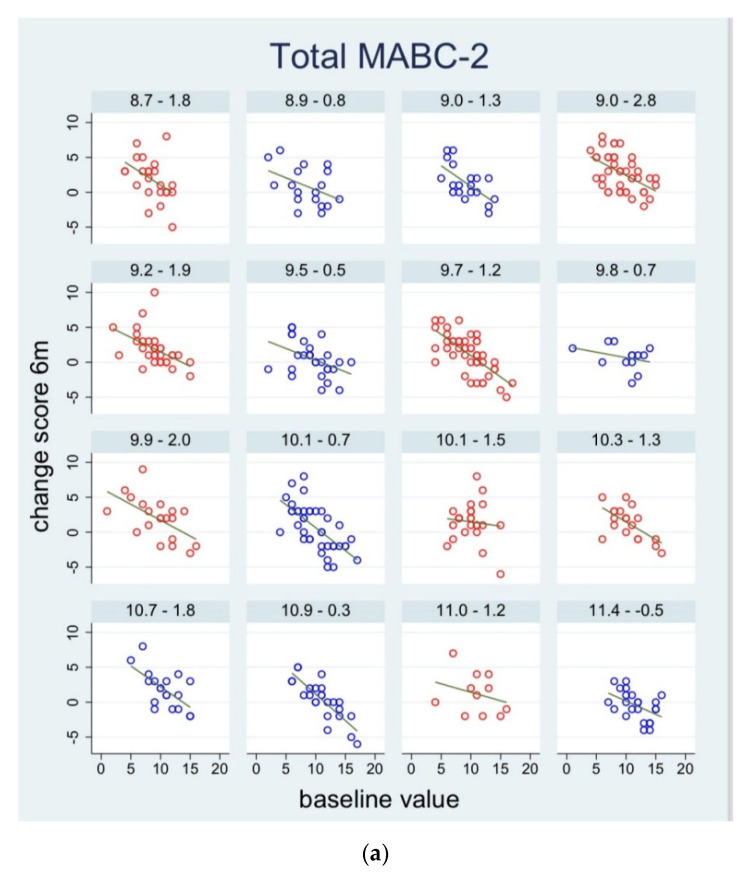
Scatter plots of change scores for the MABC-2 total score at the three follow-up times ((**a**) for 6 months follow-up, (**b**) for 18 months follow-up, and (**c**) for 30 months follow-up) vs. baseline values for each child, stratified by preschool. For each pre-school, a regression line is fitted. Preschools from the intervention group are shown in red and preschools from the control group are shown in blue. In the top of each subgraph, the mean baseline value and the mean change score are shown for each preschool. The preschools are sorted by the mean value at baseline. MABC-2: Movement Assessment Battery for Children, version 2.

**Table 1 ijerph-18-12272-t001:** At least four days a week, and preferably five, a minimum of 45 minutes adult-initiated and adult-led activities must be carried out, where all children participate. During the week, the following bodily skills must be challenged.

Bodily Skills	What and How? (Examples)	Why? (Examples)
Motor function	*Gross motor challenges* such as creeping, crawling, running, jumping, hopscotch, jumping, climbing	Gross and fine motor skills are important because they form the basis of many everyday activities and are an important factor for the child’s enjoyment of physical activity and thereby also the amount of the child’s physical expression.
	*Fine motor challenges* such as holding a pencil, handling small objects like beads and construction toys or catching insects	
	*Coordination Exercises* such as crawling exercises, cross-body movements, “Angels in the Snow”, jumping jack and throwing, gripping and kicking exercises. Other examples can be rhythm and dance.	Coordination is the foundation for a long series of specific skills such as throwing and catching and many everyday activities such as pouring water into a glass.
	Different dynamic and static *balance exercises* such as walking on a line and standing on one leg.	A good balance is, amongst other things, important in relation to avoiding falls and injuries, and affects many daily activities such as putting on clothes and shoes.
Sensing	Challenges of the following senses:	We use all of our senses to collect and process the information and experiences that we give our body and our brain. The senses are thus to control and develop our balance, coordination and motor skills and are thereby important to be able to perform both defined work routines and everyday activities.
	The *vestibular* sense is stimulated for example by rolling, turning around, doing somersaults and swinging.	
	The *tactile* sense is stimulated by touch from others, for example, in the form of massage and by touching various materials and objects of different size, shape and temperature.	
	The *kinesthetic* sense is stimulated by challenging the body’s joints, muscles and tendons in different ways, for example, by bending, stretching and pushing, lifting objects of different weights and by fast and slow movements.	
Relaxation	The children will also experience other types of physical stimulus, namely relaxation and unwinding. It can, for example, be through massage, children’s yoga or similar.	Relaxation is a good counterpart to dynamic activity, which together promote body consciousness in children. At the same time, relaxation helps to create calmer children and fewer conflicts.

**Table 2 ijerph-18-12272-t002:** Baseline description of the cohort by participants and non-participants in the RCT.

	Non-RCT	RCT	Total	*p*-Value	Missing/*n* (%)
*n* (%)	375 (44.4)	470 (55.6)	845 (100.0)		0/845 (0.00)
Age, mean years (sd)	4.39 (0.84)	4.44 (0.82)	4.41 (0.83)	0.38	1/845 (0.01)
Sex					
Boys, *n* (%)	193 (51.6)	240 (51.0)	433 (51.2)		
Girls, *n* (%)	182 (48.4)	230 (49.0)	412 (48.8)	0.85	0/845 (0.00)
Mother’s highest education:					
secondary school, *n* (%)	14 (4.5)	13 (3.2)	27 (3.8)		
low or vocational, *n* (%)	91 (29.1)	131 (32.3)	222 (30.9)		
medium or high, *n* (%)	208 (66.5)	262 (64.5)	470 (65.4)	0.52	126/845 (14.88)
Father’s highest education:					
secondary school, *n* (%)	7 (2.6)	20 (5.4)	27 (4.2)		
low or vocational, *n* (%)	98 (35.8)	154 (41.8)	252 (39.3)		
medium or high, *n* (%)	169 (61.7)	194 (52.7)	363 (56.5)	0.04	203/845 (25.86)
Equivalized income, mean DKK (sd)	315,787 (167,551)	319,686 (145,693)	318,367 (155,258)	0.76	177/845 (20.90)

RCT: Randomized controlled trial; non-RCT: participating in the cohort but not the RCT.

**Table 3 ijerph-18-12272-t003:** Drop-out analyses. Comparison of baseline values at baseline and the three follow-up samples, mean (SD).

	Baseline, *n* = 471	6 Months, *n* = 447	18 Months, *n* = 246	30 Months, *n* = 96
Age	4.44 (0.82)	4.43 (0.81)	3.91(0.53)	3.31 (0.21)
Sex (% male)	51.2	50.5	51.1	55.7
Weight (kg)	18.27 (3.07)	18.24 (2.85)	17.19 (2.67)	15.72 (1.93)
Height (cm)	106.50 (7.38)	106.50 (7.29)	102.93 (6.21)	98.06 (4.71)
Waist circumference (cm)	52.86 (3.85)	52.83 (3.58)	52.33 (3.88)	51.42 (3.41)
Total MABC-2	9.80 (3.12)	9.82 (3.12)	9.49 (2.86)	9.30 (3.01)
MABC-2 man. dex.	10.26 (3.12)	10.20 (3.18)	10.14 (2.93)	9.35 (3.26)
MABC-2 balance	10.26 (3.11)	10.26 (3.09)	10.01 (3.13)	9.82 (2.73)
MABC-2 aim/catch	9.52 (3.01)	9.50 (3.01)	9.28 (2.78)	9.10 (3.02)

MABC-2: Movement Assessment Battery for Children, version 2.

**Table 4 ijerph-18-12272-t004:** Age, sex, anthropometry, and motor skills measures at baseline and all follow-up points by intervention group (% for sex, otherwise mean).

	Baseline		6 Months		18 Months		30 Months	
	Intervention *n* = 262	Control *n* = 208	Intervention *n* = 246	Control *n* = 201	Intervention *n* = 130	Control *n* = 112	Intervention *n* = 55	Control *n* = 41
Age (years)	4.42	4.46	4.95	5.00	5.49	5.45	5.85	5.84
Sex (% male)	52.29	49.04	52.03	48.26	50.00	50.00	56.36	56.10
Weight (kg)	18.28	18.27	19.45	19.33	21.08	20.53	21.83	21.22
Height (cm)	106.38	106.64	110.31	110.74	114.43	114.17	116.90	116.44
Waist circumference (cm)	52.88	52.84	52.32	52.26	54.39	54.13	56.77	55.40
Total MABC-2	9.56	10.09	11.12	10.71	11.45	11.23	11.43	12.08
MABC-2 man. dex.	10.09	10.48	11.60	10.82	11.64	11.93	11.58	11.85
MABC-2 balance	10.27	10.26	11.61	11.04	11.50	11.44	11.21	12.59
MABC-2 aim/catch	9.22	9.89	10.09	9.88	10.66	10.09	11.28	11.61

MABC-2: Movement Assessment Battery for Children, version 2.

**Table 5 ijerph-18-12272-t005:** Change scores from baseline to follow-up by intervention group. Motor skills measured using MABC-2 in preschool children.

	6 Months			18 Months			30 Months		
	Intervention *n* = 214 *	Control *n* = 184 *	*p*-Values **	Intervention *n* = 109 *	Control *n*= 97 *	*p*-Values **	Intervention *n* = 31 *	Control *n* = 30 *	*p*-Values **
Objectively measured									
Total MABC-2	1.75 (2.74)	0.65 (2.68)	*0.004*	2.14 (2.95)	1.63 (2.86)	0.225	2.87 (3.36)	2.37 (3.33)	0.574
Manual dexterity	1.59 (3.09)	0.30 (2.92)	*0.002*	1.45 (3.19)	1.66 (3.07)	0.568	2.44 (4.08)	2.45 (3.63)	0.996
Balance	1.31 (3.23)	0.79 (3.05)	0.204	1.57 (3.45)	1.55 (3.62)	0.970	1.56 (3.81)	2.77 (3.68)	0.255
Aim/catch	0.97 (3.21)	0.02 (3.23)	0.055	1.57 (3.47)	0.42 (3.05)	*0.023*	2.78 (3.48)	1.86 (3.92)	0.334

Results presented as means with standard deviations and statistically significant *p*-values indicated with italics. * Completed total scores (individual category scores slightly higher numbers). ** *p*-values based on bivariate analyses adjusted for the cluster effect of preschools. MABC-2: Movement Assessment Battery for Children, version 2.

**Table 6 ijerph-18-12272-t006:** Effect of intervention (β-coefficient with 95% confidence interval) on motor skills in preschool children at the three follow-up times. Motor skills measured using MABC-2.

	β-Coefficient (95% CI)	*p*	β-Coefficient (95% CI)	*p*	β-Coefficient (95% CI)	*p*
All children included in the RCT	6 Months, *n* = 398 *		18 Months, *n* = 206 *		30 Months, *n* = 61 *	
Total MABC-2	0.80 (0.29; 1.31)	*0.002*	0.41 (−0.27; 1.10)	0.237	0.14 (−1.10; 1.38)	0.823
MABC-2 Manual dexterity	1.06 (0.46; 1.66)	*0.001*	−0.19 (−0.95; 0.57)	0.624	−0.13 (−0.31; 1.05)	0.829
MABC-2 Balance	0.55 (−0.09; 1.18)	0.093	0.11 (−0.71; 0.94)	0.784	−0.68 (−0.03; 1.67)	0.323
MABC-2 Aim and catch	0.49 (−0.05; 1.03)	0.073	0.84 (0.14; 1.54)	*0.019*	0.25 (−0.92; 1.42)	0.676
**Only children with risk of motor deficiency disorder, i.e., scores below the 15th percentile, included.**	**6 months, *n* = 63 ***		**18 months, *n* = 34 ***		**30 months, *n* = 13 ***	
Total MABC-2	0.41 (−0.86; 1.69)	0.527	1.42 (−0.25; 3.09)	0.096	0.79 (−1.61; 3.19)	0.517
MABC-2 Manual dexterity	0.94 (−0.72; 2.61)	0.268	−0.07 (−0.46; 2.33)	0.956	−0.01 (−0.07; 3.05)	0.996
MABC-2 Balance	0.18 (−1.38; 1.75)	0.819	0.62 (−1.18; 2.42)	0.501	−0.31 (−0.89; 3.27)	0.865
MABC-2 Aim and catch	0.18 (−1.66; 2.02)	0.849	0.54 (−1.71; 2.79)	0.639	−2.61 (−0.39; 0.17)	0.066
**Only children from preschools with mean score below the median at baseline are included in analyses, *n* = 245**	**6 months, *n* = 231 ***		**18 months, *n* = 125 ***		**30 months, *n* = 54 ***	
Total MABC-2	1.07 (0.38; 1.76)	*0.002*	1.32 (0.35; 2.30)	*0.008*	0.78 (−0.94; 2.51)	0.374
MABC-2 Manual dexterity	1.75 (1.06; 2.43)	*0.000*	−0.08 (−0.02; 0.87)	0.874	0.73 (−0.93; 2.39)	0.386
MABC-2 Balance	0.22 (−0.67; 1.10)	0.628	0.59 (−0.50; 1.68)	0.286	0.04 (−1.61; 1.70)	0.959
MABC-2 Aim and catch	0.23 (−0.69; 1.15)	0.625	1.84 (0.68; 2.99)	*0.002*	−1.01 (−0.98; 0.82)	0.267

Preschool and child included as random effects and adjusted for baseline values; statistically significant p-values indicated with italics. * Completed total scores (individual category scores slightly higher numbers).CI: Confidence interval; MABC-2: Movement Assessment Battery for Children, version 2.

## Data Availability

The data that support the findings of this study are not publicly available. Data are available upon reasonable request and with permission of the steering group.

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
