# Peer review of "The Effect of a Structured Intervention to Improve Motor Skills in Preschool Children: Results of a Randomized Controlled Trial Nested in a Cohort Study of Danish Preschool Children, the MiPS Study"

_ijerph, 2021, doi:10.3390/ijerph182312272_

Round 1

Reviewer 1 Report

The manuscript has improved with the modifications made according to the reviewers' comments. However, there are still some issues that should be improved. For example, I keep insisting that publishing first the results referring to the implementation process would give more solidity to the results referring to the effects of the intervention. The authors reply that the first manuscript regarding implementation is going through a very slow peer review process. I consider that this answer does not counteract the need to first know those results.

Referring Appendix 1: It would be necessary to provide a version in which the deleted information is crossed out in order to be able to compare the old and new versions (as is done with the main text).

There is still a need for a greater theoretical basis to justify the importance of the topic. A review of the state of the art continues to be lacking. The authors justify that: "The manuscript is already very long, and we don't agree that extending the background further is warranted". I consider that it is not a valid argument: a manuscript must be "balanced" in all its components.

The different educational levels coded from 1 to 8 remain unspecified. Only 1 (lower secondary school) and 8 (master level or above) are specified. What about 2,3,4,5,6 and 7?

The format of the Tables must be taken care of. For example, in Table 2 letters of different sizes continue to appear. In Table 4 and Table 5, in the first rows, some elements appear in bold, others do not (for example, intervention, control, ...).

In Table 5 and Table 6 different characters (bold and underlined) are used to indicate the significant p-values. They must be unified and indicate at the bottom of the table what these characters mean. Why is * not used? (it is the most usual)

It is suggested that only changes from the previous version be marked in red. In the Discussion and Conclusion everything appears in red, but not everything is new.

In Discussion Section, there is no capital letter after the period: (…) why the effect of the intervention appears to decrease over time in our study. one possible explanation is (..)

References are still not updated. The authors were already informed about it in the previous review.

The errors indicated in the References have not been corrected either. The authors reply: "We will leave potential manual corrections to the editorial office". I consider that this is not a "responsible" answer. The authors must take care of their manuscript in all the details and correct errors that are indicated.

Author Response

The manuscript has improved with the modifications made according to the reviewers' comments. However, there are still some issues that should be improved.

Point 1: For example, I keep insisting that publishing first the results referring to the implementation process would give more solidity to the results referring to the effects of the intervention. The authors reply that the first manuscript regarding implementation is going through a very slow peer review process. I consider that this answer does not counteract the need to first know those results.

Response 1: We have changed the referencing, so the two unpublished manuscripts now are referred to as other references but marked as ‘submitted, in peer review’ in the reference list. We hope this is what the reviewer wishes rather that holding this manuscript back (especially since this is for a special edition with a limited timeframe)

Point 2: Referring Appendix 1: It would be necessary to provide a version in which the deleted information is crossed out in order to be able to compare the old and new versions (as is done with the main text).

Response 2: We were not aware of the need to see the original appendix, but this is now submitted with tracked changes.

Point 3: There is still a need for a greater theoretical basis to justify the importance of the topic. A review of the state of the art continues to be lacking. The authors justify that: "The manuscript is already very long, and we don't agree that extending the background further is warranted". I consider that it is not a valid argument: a manuscript must be "balanced" in all its components.

Response 3: We have expanded the background in areas where we felt it could be improved and hope this is satisfactory.

Point 4: The different educational levels coded from 1 to 8 remain unspecified. Only 1 (lower secondary school) and 8 (master level or above) are specified. What about 2,3,4,5,6 and 7?

Response 4: All levels are now mentioned:

“Mother’s and father’s highest completed education categorized into three categories: 1) secondary school or less (early childhood education, primary education, lower secondary education, upper secondary education), 2) vocational or short education (post-secondary non-tertiary education, short-cycle tertiary education, bachelor’s or equivalent level; i.e. ≤3 years after high school), and 3) academic education (Master’s or equivalent level, post-doctoral or equivalent level, i.e. >3 years)”

Point 5: The format of the Tables must be taken care of. For example, in Table 2 letters of different sizes continue to appear. In Table 4 and Table 5, in the first rows, some elements appear in bold, others do not (for example, intervention, control, ...).

In Table 5 and Table 6 different characters (bold and underlined) are used to indicate the significant p-values. They must be unified and indicate at the bottom of the table what these characters mean. Why is * not used? (it is the most usual)

Response 5: Changed

Point 6: It is suggested that only changes from the previous version be marked in red. In the Discussion and Conclusion everything appears in red, but not everything is new.

Response 6: Since large parts of the discussion and conclusion have been rewritten, the resulting track-changes version is not readable. However, the original version of the discussion and conclusion was included in the revised manuscript for referral

Point 7: In Discussion Section, there is no capital letter after the period: (…) why the effect of the intervention appears to decrease over time in our study. one possible explanation is (..)

Response 8: Changed

Point 8: References are still not updated. The authors were already informed about it in the previous review.

The errors indicated in the References have not been corrected either. The authors reply: "We will leave potential manual corrections to the editorial office". I consider that this is not a "responsible" answer. The authors must take care of their manuscript in all the details and correct errors that are indicated.

Response 8: Since references are formatted electronically in Endnote, we have corresponded with the editorial office about this, and they state that “We will have a professional team to modify the format of the references.” Therefore, we have not taken any action on this point.

Reviewer 2 Report

Editorial corrections increased the transparency of the work for the reader. The authors expanded the discussions significantly. They attempted to explain the research results, but unfortunately they mainly referred to social or organizational aspects. There was no attempt to explain the results in terms of developmental and physiological aspects.

Author Response

Point 1: Editorial corrections increased the transparency of the work for the reader. The authors expanded the discussions significantly. They attempted to explain the research results, but unfortunately they mainly referred to social or organizational aspects. There was no attempt to explain the results in terms of developmental and physiological aspects.

Response 1: It is true, that we mainly focus on social and organizational aspects in our discussion. This is because we find these explanations most likely to explain the change over time. However, we do think there is an upper limit to the degree of improvement based on developmental and physiological aspects which is what we refer to in the following and other sentences: “Another possible explanation could be that the intervention accelerated the development of the children’s motor skills until they reached a certain degree of saturation, whereafter the control group caught up later, just at a slower pace.” To make that point more clear, we have changed the sentence to: “Another possible explanation could be that the intervention accelerated the development of the children’s motor skills until they reached their developmental potential, whereafter the control group caught up later, just at a slower pace.”

This manuscript is a resubmission of an earlier submission. The following is a list of the peer review reports and author responses from that submission.

Round 1

Reviewer 1 Report

The article is extensive, methodologically correct, the results are presented very clearly. I evaluate the discussion of the results low.

Background

It is accurate. It shows the topic of children's motor skills from many important points of view: health, physical activity, future development.

  • „There are indications that the level of motor skills remains stable over time [17] and motor development deficits observed in early childhood are still apparent in adolescence”.
    The first part of this sentence does not seem to be correct. Rather, we are not saying that motor skills are constant but we note that deficits in motor skills influence further development. The authors themselves note in the second part of this sentence: motor development deficits observed in early childhood are still apparent in adolescence. Moreover, the first part of this sentence was supported by a 1984 publication which is a very ancient report. It is worth considering this sentence

Methods

This part is described in great detail.

  • As Authors wrote: „the intervention was not a strictly defined curriculum, but a strategy to enhance motor skills during the preschool day while fulfilling defined minimum criteria”. Whether there was a detailed record of activities undertaken in individual institutions?
  • How the intervention and control groups were selected?

Results

  • Figure 1. is not clear. „n = 845” is the total number of invitees to participate? In the text Authors wrote „Of the 834 children invited to participate”. Moreover in Table 1. the number of all participants is 847. Please verify.
  • The abbreviations used in Figure 1. need to be elaborated on in the legend.

Discussion

  • In my opinion, there was no attempt to answer the key question: is intervention necessary?
  • A temporary improvement of the measured variables was shown, why?
  • The first paragraph of the discussion repeats the results.
  • In the discussion, the authors refer the results of their own research only to Australian reports.

Reviewer 2 Report

Comments and suggestions are in the the attached file.
